# First Assessment of the Earth Heat Inventory Within CMIP5 Historical Simulations

Francisco José Cuesta-Valero[1,2], Almudena García-García[1,3], Hugo Beltrami[1], and Joel Finnis[4]

[1]Climate & Atmospheric Sciences Institute, St. Francis Xavier University, Antigonish, NS, Canada.
[2]Environmental Sciences Program, Memorial University of Newfoundland, St. John's, NL, Canada.
[3]Department of Remote Sensing, Helmholtz Centre for Environmental Research-UFZ, Leipzig, Germany.
[4]Department of Geography, Memorial University of Newfoundland, St. John's, NL, Canada.

**Correspondence:** Hugo Beltrami (hugo@stfx.ca)

**Abstract.** The energy imbalance at the top of the atmosphere over the last century has caused an accumulation of heat within the ocean, the continental subsurface, the atmosphere and the cryosphere. Although $\sim$90% of the energy gained by the climate system has been stored in the ocean, the other components of the Earth heat inventory cannot be neglected due to their influence on associated climate processes dependent on heat storage, such as sea level rise and permafrost stability. However, there has

not been a comprehensive assessment of the heat inventory within global climate simulations yet. Here, we explore the ability of thirty advanced General Circulation Models (GCMs) from the fifth phase of the Coupled Model Intercomparison Project (CMIP5) to simulate the distribution of heat within the Earth's energy reservoirs for the period 1972-2005 of the Common Era. CMIP5 GCMs simulate an average heat storage of $247 \pm 172$ ZJ ($96 \pm 4$ % of total heat content) in the ocean, $5 \pm 9$ ZJ ($2 \pm 3$ %) in the continental subsurface, $2 \pm 3$ ZJ ($1 \pm 1$ %) in the cryosphere, and $2 \pm 2$ ZJ ($1 \pm 1$ %) in the atmosphere.

However, the CMIP5 ensemble overestimates the ocean heat content by $83$ ZJ and underestimates the continental heat storage by $9$ ZJ and the cryosphere heat content by $5$ ZJ, in comparison with recent observations. The representation of terrestrial ice masses and the continental subsurface, as well as the response of each model to the external forcing, should be improved in order to obtain better representations of the Earth heat inventory and the partition of heat among climate subsystems in global transient climate simulations.

## 1 Introduction

Sustained net radiative imbalance at the top of the atmosphere is increasing the heat stored within the climate subsystems –the ocean, the continental subsurface, the atmosphere and the cryosphere (Hansen et al., 2011; von Schuckmann et al., 2020). The ocean is the largest component of the Earth Heat Inventory (EHI), accounting for around 90% of the total heat in the climate system (Rhein et al., 2013; Gleckler et al., 2016; von Schuckmann et al., 2020). Nonetheless, it is imperative to measure the

distribution of heat storage within the four components of the climate system, since the evolution of several physical processes critical to understand climate change and quantify future impacts of climate change on society are strongly dependent on the partition of heat among all climate components.

The evolution of ocean heat content determines the thermosteric component of sea level rise (Church et al., 2011; Kuhlbrodt and Gregory, 2012; Levitus et al., 2012), affects the total precipitation and intensity of hurricanes (Mainelli et al., 2008; Wada and Chan, 2008; Lin et al., 2013; Trenberth et al., 2018), and influences regional cyclonic activity (Bhowmick et al., 2016). The increase in ground heat content leads to the warming of the continental subsurface and to permafrost thawing in the Northern Hemisphere (Koven et al., 2013; Cuesta-Valero et al., 2016; McGuire et al., 2018; Biskaborn et al., 2019; Hock et al., 2019; Meredith et al., 2019; Soong et al., 2020). Thus, the increase in continental heat storage threatens the stability of the global soil carbon pool, potentially facilitating the release of large amounts of greenhouse gasses from the decomposition of soil organic matter in northern soils (Koven et al., 2011; MacDougall et al., 2012; Schädel et al., 2014; Schuur et al., 2015; Hicks Pries et al., 2017; McGuire et al., 2018). Melting of ice sheets in Greenland and Antarctica as well as glacier degradation at all latitudes contribute to sea level rise (Jacob et al., 2012; Hanna et al., 2013; Vaughan et al., 2013; Dutton et al., 2015; Bamber et al., 2018; King et al., 2018; Rignot et al., 2019; Hock et al., 2019; Oppenheimer et al., 2019; Meredith et al., 2019; Zemp et al., 2019), and together with changes in the extension and volume of sea ice may disturb deep water formation zones and alter ocean circulation and large scale heat distribution (Hu et al., 2013; Jahn and Holland, 2013; Ferrari et al., 2014; Smeed et al., 2018; Collins et al., 2019). The evolution of the atmosphere heat content constrains the projected change in total global precipitation due to atmospheric warming (Pendergrass and Hartmann, 2014a; Hegerl et al., 2015), and the additional moisture in a warmer atmosphere increases the frequency of extreme precipitation events (Pendergrass and Hartmann, 2014b). The intensity of cyclones and hurricanes is also expected to increase in the future due to the higher energy available in the atmosphere (Pan et al., 2017).

Therefore, the partition of heat within these subsystems have long-term impacts on society, as the heat content of each subsystem is related to processes altering near-surface conditions. Higher surface temperatures together with changes in precipitation regimes and sea level rise threaten global food security (Lloyd et al., 2011; Rosenzweig et al., 2014; Phalkey et al., 2015; Campbell et al., 2016), and may result in an increase in the frequency of floods and storm surges (McGranahan et al., 2007; Lin et al., 2013; Kundzewicz et al., 2014). The combination of high temperature, high levels of moisture, and changes in precipitation patterns also affect human health, particularly for the populations least responsible for climate change (Patz et al., 2007). These changes in near-surface conditions increase the risk of high levels of heat stress (Sherwood and Huber, 2010; Matthews et al., 2017) and the spread of infectious diseases (Levy et al., 2016; Wu et al., 2016; McPherson et al., 2017), among others risks for human health (McMichael et al., 2006).

General Circulation Model (GCM) simulations are the main source of information about the possible evolution of the climate system, which is critical for society's adaptation to future risks posed by climate change. Modeling experiments performed for the fifth phase of the Coupled Model Intercomparison Project (CMIP5) have provided several insights into the long-term evolution of the net radiative imbalance at the top of the atmosphere (Allan et al., 2014; Smith et al., 2015), the evolution of ocean heat content since preindustrial times (Gleckler et al., 2016), and the relationship between these two magnitudes (Palmer et al., 2011; Palmer and McNeall, 2014; Smith et al., 2015). The same GCM simulations, nevertheless, do not simulate other aspects of the Earth heat inventory successfully. CMIP5 simulations are unable to accurately represent heat storage within the continental subsurface over the second half of the 20[th] century (Cuesta-Valero et al., 2016), many do not conserve atmospheric

water (Liepert and Lo, 2013) nor subsurface water (Krakauer et al., 2013; Trenberth et al., 2016), and do not conserve the total heat in the system (Hobbs et al., 2016). Furthermore, there has not yet been an assessment of the ability of CMIP5 GCMs to reproduce heat storage within the atmosphere and the cryosphere, despite their impact on a variety of phenomena of critical interest to both society and the scientific community.

Here, we assess the ability of thirty CMIP5 GCM Historical simulations to reproduce the Earth heat inventory and the partition of heat within the ocean, continental subsurface, atmosphere and cryosphere. Results are compared with observations for the period 1972-2005 of the Common Era (CE). Our analysis reveals the importance of the simulated terrestrial ice masses and the represented continental subsurface volume for achieving a realistic distribution of the total Earth heat content within GCM simulations, and reinforces the need to reduce the spread in model responses to external forcing.

## 2 Data and Methods

Thirty Historical simulations performed with advanced general circulation models were retrieved from the fifth phase of the Coupled Model Intercomparison Project (CMIP5) archive (Taylor et al., 2011). Historical simulations attempt to represent the evolution of global climate from the Industrial Revolution to the present (1850-2005 CE) using estimates of natural and anthropogenic emissions of greenhouse gases and aerosols, as well as changes in land cover and land use (Mieville et al., 2010; Hurtt et al., 2011). We analyzed the simulated evolution of heat storage in the entire climate system and in the different subsystems (the ocean, continental subsurface, atmosphere and cryosphere) for the period 1972-2005 CE, in common with observations.

Estimates of the Earth heat inventory from observations are retrieved from Church et al. (2011) and von Schuckmann et al. (2020). Results in Church et al. (2011) are provided for the period 1972-2008 CE, thus we scale those estimates linearly to cover the period 1972-2005 CE, in common with CMIP5 Historical simulations. von Schuckmann et al. (2020) provides observational estimates from 1960 to 2018 CE at annual resolution, thus results for the period 1972-2005 CE are selected without scaling or modification. Both datasets employ similar measurements from mechanical and expendable bathythermographs to estimate the heat content within the ocean. Differences in the reported heat storage are caused by the statistical treatment of data gaps, the choice of the climatology, the approach to account for instrumental biases, and the higher number of recent measurements included in von Schuckmann et al. (2020). Church et al. (2011) extrapolates the continental heat storage estimated in Huang (2006) from meteorological observations of surface air temperature at $2$ m. Otherwise, von Schuckmann et al. (2020) includes ground heat content estimates from an updated database of borehole temperature profile measurements (Cuesta-Valero et al., 2021). This method contrasts to the one included in Church et al. (2011), since estimates of continental heat storage are retrieved from direct measurements of subsurface temperatures. There are substantial differences between both datasets in the methods employed to obtain the heat storage in the atmosphere. Church et al. (2011) estimates heat storage as proportional to the change in surface air temperature, while von Schuckmann et al. (2020) considers the atmospheric profile in several reanalysis products, multisatellite radio occultation records, and radiosonde observations (Steiner et al., 2020), analyzing temperature, water content and wind intensity. Estimates of ice melting from glaciers and ice sheets are considered in both datasets, with more recent

analyses included in von Schuckmann et al. (2020). Changes in sea ice volume in Church et al. (2011) are obtained from Levitus et al. (2005), and from the Pan-Arctic Ice Ocean Modeling and Assimilation System (PIOMAS, Zhang and Rothrock, 2003; Schweiger et al., 2019) in the case of von Schuckmann et al. (2020). All changes in ice mass are multiplied by the latent heat of fusion in order to obtain the corresponding estimate of cryosphere heat content.

Global averages of Ocean Heat Content (OHC), the heat content within the continental subsurface (ground heat content, GHC), Atmosphere Heat Content (AHC) and heat uptake by ice masses (cryosphere heat content, CHC) were derived from the CMIP5 Historical experiments. The OHC values were estimated using the formulation for potential enthalpy described in McDougall (2003) and Griffies (2004) from simulated seawater potential temperature and salinity profiles (Table 1 contains the list of variables employed for estimating each term of the EHI). Once the potential enthalpy has been determined, estimates

of seawater density (McDougall et al., 2003) and pressure profiles (Smith et al., 2010) allowed simulated heat content in the ocean to be calculated as:

$$Q_{Ocean} = \sum_{i=z_0}^{z_f} \rho_i\left(S, \theta, p(z_i)\right) \cdot \mathcal{H}_i^{\circ}\left(S, \theta\right), \cdot \Delta z_i, \tag{1}$$

where $Q_{Ocean}$ is the ocean heat per surface unit (in $\mathrm{J\,m^{-2}}$), $S$ is salinity (in psu), $\theta$ is potential temperature (in $^{\circ}$C), $p$ is pressure (in bar), and $z_i$, $\rho_i$, $\mathcal{H}_i^{\circ}$ and $\Delta z_i$ are depth (in m), density (in $\mathrm{kg\,m^{-3}}$, potential enthalpy (in $\mathrm{J\,kg^{-1}}$) and thickness (in

105    m) of the i-th ocean layer, respectively. This approach is based on the availability of both temperature and salinity profiles in CMIP5 simulations, which allows to integrate changes in water density. Estimates of OHC from observational methods only consider temperature profiles, as salinity profiles are not routinely measured at the global scale. However, CMIP5 simulations yield similar changes in OHC from both methods (Figure S1). Thus, we use the method described by Equation 1 to estimate OHC from simulations, since this approach includes simulated salinity profiles in the analysis, maximizing the information

considered to estimate heat content.

The GHC series were estimated as in Cuesta-Valero et al. (2016) for all terrestrial grid cells. Subsurface thermal properties were computed taking into account spatial variations in soil composition (% of sand, clay and bedrock) and simulated subsurface water and ice amounts (Van Wijk et al., 1963; Oleson et al., 2010). The subsurface temperature profile was then integrated following

$$Q_{Ground} = \sum_{i=z_0}^{z_f} \rho C_i \cdot T_i \cdot \Delta z_i, \tag{2}$$

where $Q_{Ground}$ is the subsurface heat storage per surface unit (in $\mathrm{J\,m^{-2}}$), and $\rho C_i$, $T_i$ and $\Delta z_i$ are the volumetric heat capacity (in $\mathrm{J\,m^{-3}\,K^{-1}}$), the temperature (in K) and the thickness (in m) of the i-th soil layer, respectively. All CMIP5 GCMs present outputs for subsurface temperature, but not all models provide outputs for subsurface water and ice content in the same format (Table 1), hampering the estimate of thermal properties ($\rho C$) in Equation 2. Indeed, two thirds of the GCMs provide the joint

content of water and ice for each soil layer (*mrlsl* variable in CMIP5 notation), while the remaining third provide the total water and ice content in the entire soil column (*mrso* variable). As in Cuesta-Valero et al. (2016), we considered water to be frozen in layers with temperatures below 0 $^{\circ}$C and liquid water otherwise for models providing the *mrlsl* variable. For models

providing the *mrso* variable, we distributed the water and ice content among the soil layers proportionally with layer thickness, considering ice in soil layers with temperature below 0 °C and liquid water otherwise.

The AHC series from CMIP5 simulations were estimated using the theoretical foundations of Trenberth (1997) and Previdi et al. (2015). The simulated air temperature profile was integrated for all atmospheric grid cells together with estimates of wind kinetic energy, latent heat of vaporization and surface geopotential, which was determined as in Dutton (2002). Vertical atmospheric profiles were integrated in pressure coordinates as:

$$Q_{Atmosphere} = \frac{1}{g} \sum_{i=0}^{p_s} (c_p \cdot T_i + k_i + L \cdot q_i + \Phi_s) \Delta p_i, \tag{3}$$

where $Q_{Atmosphere}$ is atmospheric heat per surface unit (in $\mathrm{J\,m^{-2}}$), $g$ is apparent acceleration due to gravity (in $\mathrm{m\,s^{-2}}$), $p_s$ is surface pressure (in Pa), $c_p = 1000 \ \mathrm{J\,kg^{-1}\,K^{-1}}$ is the specific heat of air at constant pressure, $L = 2260 \ \mathrm{J\,kg^{-1}}$ is the latent heat of vaporization, $\Phi_s$ is the surface geopotential estimated from orography (in $\mathrm{m^2\,s^{-2}}$), and $T_i$, $k_i$, $q_i$ and $\Delta p_i$ are the air temperature (in K), specific kinetic energy (in $\mathrm{J\,kg}$), specific humidity (in $\mathrm{kg\,kg^{-1}}$) and thickness (in Pa) of the i-th atmospheric layer, respectively.

For estimating the CHC series, the simulated cryosphere was divided into three terms: sea ice, subsurface ice and glaciers. Variations in the mass of simulated sea ice and subsurface ice were multiplied by the latent heat of fusion ($L_f = 3.34 \times 10^5 \ \mathrm{J\,kg^{-1}}$, Rhein et al., 2013) to obtain the heat absorbed in the melting process. The same method was applied to the change in snow mass in grid cells containing land ice within each CMIP5 GCM (glaciers or ice sheets, *sftgif* variable in the CMIP5 archive). Although this is not a satisfactory approach given the differences between snow and land ice, it is the only
available approximation since CMIP5 GCMs do not typically represent terrestrial ice masses (Flato et al., 2013). Therefore, the cryosphere heat content was estimated as

$$Q_{Cryosphere} = L_f \cdot (\Delta \omega + \rho \cdot \Delta p \cdot \Delta z + \Delta \Omega), \tag{4}$$

where $Q_{Cryosphere}$ is absorbed heat per surface unit (in $\mathrm{J\,m^{-2}}$), $\rho = 920 \ \mathrm{kg\,m^{-3}}$ is ice density (Rhein et al., 2013), $\Delta \omega$ is the change in subsurface ice mass per surface unit (in $\mathrm{kg\,m^{-2}}$), $\Delta p$ is the change in the proportion of sea ice at each ocean grid
cell, $\Delta z$ is the change in thickness of sea ice at each ocean grid cell (in m), and $\Delta \Omega$ is the change in snow amount at each cell containing land ice (in $\mathrm{kg\,m^{-2}}$). It is important to note that nine of the CMIP5 GCMs did not provide outputs for the subsurface ice amount (*mrfso* variable) and that three of the models did not provide outputs for snow amount (*snw* variable, see Table 1), and thus these terms are missing in the CHC estimates from those models. We were unable to retrieve the file indicating the cells containing land ice (*sftgif* file) for the HADGEM2-CC GCM, thus we used the CMCC-CMS *sftgif* file interpolated to the
HADGEM2-CC grid, since the grid for both models have a similar spatial resolution (1.25 ° × 1.875 ° for HADGEM2-CC; 1.875 ° × 1.875 ° for CMCC-CMS).

Estimates of total heat in the climate system from each CMIP5 model are required to determine the simulated partition of heat among each climate subsystem. The total heat content can be determined as the sum of the heat storage within the different climate subsystems (Earth heat content, EHC) or as the integration of the radiative imbalance at the top of the atmosphere (N)
during the period of interest. Both approximations have been used in the literature and are considered equivalent (Rhein et al.,

2013; Palmer and McNeall, 2014; Trenberth et al., 2014; von Schuckmann et al., 2016). That is, if a model does not produce artificial sources or leakages of energy or mass (i.e., if the model conserves the total heat content in the system), the change in N and in EHC should be almost identical (Hobbs et al., 2016). Nevertheless, CMIP5 GCM simulations are prone to drift, particularly the ocean component due to incomplete model spin-up procedures (Sen Gupta et al., 2013; Séférian et al., 2016).

For this reason, potential drifts in estimates of heat content and the components of the radiative budget at the top of the atmosphere were removed by subtracting the linear trend of the corresponding preindustrial control simulation from the Historical simulations, which should correct artificial drifts in the simulated heat content within each climate subsystem (Hobbs et al., 2016). N estimates from the CESM1-CAM5 GCM constitute a particular case, since an unrealistic trend remained in the Historical experiment in comparison with other CMIP5 GCMs after removing the drift using data from the corresponding control

simulation (Figure S2). The rest of variables from this GCM were dedrifted using the trend estimated from the preindustrial control simulation as in the other CMIP5 simulations, but the drift in the outgoing shortwave radiation and the outgoing longwave radiation at the top of the atmosphere could not be removed. Therefore, we used the trend estimated from the first five decades of the Historical simulation (1861-1911 CE) to remove the drift in N estimates, achieving a better comparison with the other CMIP5 GCMs (Figure S2).

As a complement to the estimates of the EHI detailed above, we also estimated the partition of the simulated total heat content among the ocean, the continental subsurface, the atmosphere and the cryosphere. A linear regression analysis was performed between the evolution of the simulated heat storage within each climate subsystem and the estimates of total heat content in the entire climate system to determine the partition of heat within the four climate subsystems (Figure 1). The slope of the linear fit was assumed to represent the simulated proportion of heat in the corresponding subsystem, thus providing estimates of OHC/N

and OHC/EHC for the simulated proportion of heat in the ocean, GHC/N and GHC/EHC for the simulated proportion of heat in the continental subsurface, AHC/N and AHC/EHC for the simulated proportion of heat in the atmosphere, and CHC/N and CHC/EHC for the simulated proportion of heat absorbed by the cryosphere.

## 3    Results

### 3.1    Earth Heat Inventory

The CMIP5 ensemble mean overestimates the observed ocean heat content for the period 1972-2005 CE and underestimates the observations for the continental subsurface and the cryosphere (Figure 2). Additionally, the multimodel mean yields higher total heat in the climate system than observations, as expected due to the high OHC values reached by these simulations (Figure 2a). Indeed, the CMIP5 multimodel mean yields an OHC increase of $247 \pm 172$ ZJ (mean $\pm$ two standard deviations, $1\,\text{ZJ} = 1 \times 10^{21}$ J) for 1972-2005 CE, higher than the observational estimates in Church et al. (2011) ($\sim 199$ ZJ) and von

Schuckmann et al. (2020) ($164 \pm 17$ ZJ, Table 2). These high OHC estimates are the cause of the large Earth heat content displayed by the CMIP5 ensemble, since the EHC estimates result from the cumulative heat storage in the four climate subsystems, and the ocean accounts for around 90% of the total heat storage (Church et al., 2011; Hansen et al., 2011; Rhein et al., 2013; Gleckler et al., 2016; von Schuckmann et al., 2020). The integration of the radiative imbalance at the top of the

atmosphere for the period 1972-2005 CE should yield similar values to those of EHC and OHC over the same period, as the radiative imbalance causes the heat storage within the different climate subsystems. Indeed, EHC and OHC estimates are generally similar within each model, while N values diverge from those for the Earth heat content in some models, which may suggest that those models have biases in their represented energy budget. Particularly, the CESM1-CAM5, CMCC-CM, GFDL-CM3, HADGEM2-CC, INM-CM4, IPSL-CM5A-LR, IPSL-CM5A-MR, IPSL-CM5B-LR, NOR-ESM1-M, NOR-ESM1-ME models show N-EHC differences larger than 10% of their simulated changes in OHC (Figure 2a). Furthermore, the inter-model spread obtained for these three magnitudes is excessively large, given that all Historical simulations were forced using the same boundary conditions –i.e., the same external forcing. Further details about the large spread among the CMIP5 simulations, as well as the discrepancies in N, EHC and OHC can be found in the Discussion section.

A different situation is found for the magnitude of the simulated heat storage within the continental subsurface, with the CMIP5 ensemble mean yielding generally lower estimates of GHC than the observations (Figure 2b). The multimodel mean achieves a GHC change of $5 \pm 9$ ZJ for 1972-2005 CE, which is lower than the $14 \pm 3$ ZJ in von Schuckmann et al. (2020) but similar to the $\sim 4$ ZJ in Church et al. (2011) (Table 2). However, the difference between the GHC estimates in Church et al. (2011) and in von Schuckmann et al. (2020) is large (Figure 2), probably caused by the different source of data used in both products. That is, results from Church et al. (2011) are based on surface air temperatures while results from von Schuckmann et al. (2020) are based on subsurface temperatures (see Huang, 2006; Cuesta-Valero et al., 2021, and the Data and Methods section for more details). Therefore, the estimate of $14 \pm 3$ ZJ from von Schuckmann et al. (2020) constitutes a more robust reference for evaluating the simulated ground heat content by the CMIP5 ensemble, indicating that models underestimate observations of continental heat storage. Additionally, the representation of GHC in the CMIP5 GCMs is markedly limited by the simulated subsurface volume, which is determined by the depth of the Land Surface Model (LSM) component (Stevens et al., 2007; MacDougall et al., 2008; Cuesta-Valero et al., 2016). Indeed, five of seven GCMs using LSM components deeper than $40$ m yield GHC estimates in agreement with the 95% confidence interval of observations from von Schuckmann et al. (2020), suggesting that the underestimated continental heat storage and the large spread in the CMIP5 ensemble are direct consequences of the different bottom boundary depths used by each model (see Cuesta-Valero et al., 2016, for a complete list of bottom boundary depths). The negative GHC estimates for both MRI simulations in Figure 2b are caused by an unrealistic and sharp decrease of the total water content in the subsurface along these Historical simulations (see Cuesta-Valero et al., 2016, for more details).

The CMIP5 ensemble mean constantly underestimates the cryosphere heat content in comparison with observations (Figure 2b). The multimodel average estimates a $2 \pm 3$ ZJ change in the cryosphere heat content for the period 1972-2005 CE, which is much lower than the observed CHC in Church et al. (2011) (7 ZJ) and in von Schuckmann et al. (2020) ($7 \pm 1$ ZJ, Table 2). Figure 3 examines the three components contributing to the cryosphere heat content in this analysis for each CMIP5 model (i.e, sea ice, subsurface ice and glaciers), in order to understand the reason for the disagreement between simulated and observed CHC estimates. The simulated heat uptake to reduce sea ice volume is in agreement with observations, with a multimodel mean of $2 \pm 2$ ZJ while observations reach $\sim 2$ ZJ and $2.5 \pm 0.2$ ZJ in Church et al. (2011) and von Schuckmann et al. (2020), respectively (Figure 3, Table 2). However, the spread in the CMIP5 results is still large, with the difference between the highest

and the lowest estimates of heat storage due to sea ice melting being more than double the value of the ensemble mean (5 ZJ).

Heat uptake by subsurface ice is the second contributor to the cryosphere heat content in all models after sea ice melting. Nevertheless, neither Church et al. (2011) nor von Schuckmann et al. (2020) include observations of the change in terrestrial subsurface ice, and not all CMIP5 GCMs include a representation of the subsurface ice masses, thus we cannot assess the ability of the CMIP5 GCMs to reproduce this term of the cryosphere heat content. Furthermore, the approximation used in this study to estimate the simulated heat absorbed by glaciers yields a much smaller value from models than from observations ($\sim 2.8$ ZJ

in Church et al. (2011) and $\sim 1.4$ ZJ in von Schuckmann et al. (2020)), indicating that a comprehensive representation of terrestrial ice masses is necessary to reproduce observations.

The heat storage within the atmosphere yields the best results for the CMIP5 GCMs in comparison with observations (Figure 2b). The CMIP5 ensemble mean achieves an atmosphere heat content of $2 \pm 2$ ZJ, in agreement with observations from Church et al. (2011) (2 ZJ) and von Schuckmann et al. (2020) ($2.2 \pm 0.3$ ZJ). Additionally, one third of the models

displays AHC estimates within the 95% confidence interval of the observed atmosphere heat content. Despite the similarity between the multimodel mean and observations, the inter-model spread is large, with the difference between the maximum and minimum AHC from CMIP5 models reaching 5 ZJ, more than double the value of the observational estimate.

### 3.2 Heat Partition Within Climate Subsystems

The simulated heat storage within each climate subsystem has been assessed in the previous section, displaying a large inter-

240 model spread among CMIP5 GCMs. This wide range of results hampers the assessment of the simulated Earth heat inventory, particularly the evaluation of the represented ocean heat content and total heat in the climate system. Nevertheless, models may be distributing the total heat content among the four climate subsystems similarly. This section evaluates the partition of heat among climate subsystems within each CMIP5 GCM, testing whether models simulating higher values of N and EHC distribute this energy in the same proportion among climate subsystems as models simulating lower values of total heat content.

The simulated heat partitions by the thirty CMIP5 GCMs achieve a lower inter-model spread in comparison with the simulated EHI, particularly for the ocean component (Figures 4 and 5, Table 2). Nevertheless, the ensemble mean presents a partition of heat in each climate subsystem similar to the results for the EHI. That is, the simulated proportion of energy in the ocean is larger than observations, the proportion of heat in the continental subsurface and in the cryosphere is lower than observations, and the proportion of heat in the atmosphere is in agreement with observations. Additionally, results vary depending on the

metric used to characterize total heat content in the system, particularly for the ocean.

All thirty CMIP5 GCM simulations represent a proportion of heat stored in the ocean within the 95% confidence interval of the observations considering EHC as metric for total energy in the climate system (OHC/EHC, blue dots in Figure 4a), achieving a multimodel mean just 2% higher than Church et al. (2011) and 8% higher than von Schuckmann et al. (2020) (Table 2). The spread of OHC/EHC estimates is small, with values ranging from $91 \pm 2$ % (MIROC5) to $100 \pm 1$ % (MRI-CGCM3).

Nevertheless, the simulated proportion of heat in the ocean presents different results for some models when considering the integration of the radiative imbalance at the top of the atmosphere as metric for total heat in the climate system (OHC/N, black dots in Figure 4a). The model spread is much larger for OHC/N estimates than for OHC/EHC estimates, ranging from

56 ± 2 % (CMCC-CM) to 122 ± 4 % (NOR-ESM1-M). These different estimates are related to the differences between N and EHC values displayed in Figure 2a. That is, some CMIP5 models yield excessively different values of N and EHC, suggesting the presence of non-conservation terms in the simulated energy budget (see Section 3.1 and Discussion section). Six models obtain OHC/N estimates above 100%, which indicates that the simulated N in those models is much lower than EHC estimates (the BCC-CSM1.1-M, CANESM2, CMCC-CMS, MIROC-ESM, NOR-ESM-M and NOR-ESM-ME models in Figure 4a). The opposite behavior occurs in other five models that simulate OHC/N values below 80% (the CESM1-CAM5, CMCC-CM, GFDL-CM3, IPSL-CM5A-LR, and IPSL-CM5B-LR models in Figure 4a), which is probably a excessively small proportion of heat stored in the ocean in comparison with observations (Hansen et al., 2011; Palmer et al., 2011; Rhein et al., 2013; Palmer and McNeall, 2014; Trenberth et al., 2014; Hobbs et al., 2016; von Schuckmann et al., 2016, 2020).

Estimates of the proportion of heat in the ground from CMIP5 GCMs show smaller differences between GHC/N and GHC/EHC than the retrieved proportion of heat in the ocean (Figure 4b). Both GHC/N and GHC/EHC estimates have a multimodel mean and 95% confidence interval of 2 ± 3 %, which is in agreement with estimates derived from Church et al. (2011) (∼ 2 %), but excessively low in comparison with results from von Schuckmann et al. (2020) (7 ± 2 %). As in the case of the simulated ground heat content, the relatively large inter-model spread in the simulated proportion of heat stored in the continental subsurface is caused by the depth of the LSM component. Indeed, deeper models reach higher proportions of heat in the ground than shallower models using either EHC or N as metric for total heat in the climate system. This marked dependence on the depth of the represented subsurface is apparent in a covariance analysis, with significant correlation coefficients between the depth of the LSM component and the GHC/N and GHC/EHC estimates (Figure S3).

As in the case of the continental subsurface, CMIP5 GCMs consistently underestimate the observed proportion of heat absorbed by the cryosphere. Both metrics of total heat content in the system yield similar ratios (CHC/N and CHC/EHC), with only one model (the HADGEM2-CC) reaching the 95% confidence interval from von Schuckmann et al. (2020) (Figure 5). This disagreement between observations and CMIP5 simulations is expected given the large differences in the simulated and observed cryosphere heat content (Figure 2b), while the partial agreement between the HADGEM2-CC estimates and the observations is likely the result of the low EHC and N values simulated by this model (Figure 2a). Nevertheless, CMIP5 models and observations agree if considering only the heat allocated for sea ice melting (Table 2), with the multimodel average yielding an average of 1 ± 1 % in comparison with 1 % from Church et al. (2011) and 1.2 ± 0.2 % from von Schuckmann et al. (2020).

The CMIP5 GCMs also show similar estimates for the proportion of heat in the atmosphere using both EHC and N metrics. A large proportion of the models achieve AHC/N and AHC/EHC ratios within the 95% confidence interval from von Schuckmann et al. (2020), and contain the observational estimates from Church et al. (2011) within the limits of their individual confidence intervals (Figure 5). The ensemble average yields a proportion of heat in the atmosphere of around 1 ± 1 %, with observations reporting 0.9% (Church et al., 2011) and 1.1 ± 0.2 %, which is a reassuring result for the CMIP5 models (von Schuckmann et al. (2020), Table 2).

## 4 Discussion

The thirty CMIP5 GCMs analyzed here simulate markedly different total heat contents within the climate system, independently of the analyzed metric (N, EHC and OHC values in Figure 2a), which may be caused by the different response from each model to the common Historical forcing. That is, different models simulate distinct responses to the common external forcing, as seen in the broad range of simulated equilibrium climate sensitivities in the literature (e.g. Knutti et al., 2017). Indeed, Forster et al. (2013) assessed the response to the common forcing of a large ensemble of CMIP5 GCMs in terms of climate sensitivity, feedbacks and adjusted radiative forcing, showing that these models yielded a broad range of responses. To test the potential relationship between total heat storage and model response, we performed a covariance analysis between some of the metrics used by Forster et al. (2013) to characterize the response of CMIP5 models and the estimated Earth heat inventory here (Figure 6). The eighteen CMIP5 models in common with those analyzed in Forster et al. (2013) do not show covariance between the heat storage within the different climate subsystems and equilibrium climate sensitivity nor with the transient climate response. However, the adjusted forcing during the last part of the Historical experiment (2001-2005 CE) presents significant correlation coefficients with N, EHC and OHC (red triangles in Figure 6). This is a reasonable result, as different adjusted forcings result from a spread of radiative imbalances at the top of the atmosphere and climate sensitivities, from which different N values arise –and therefore distinct heat storage within the ocean (Palmer and McNeall, 2014). The relationship between adjusted forcing and heat storage, nevertheless, should be considered just as a potential line of research, since the estimates of radiative forcing from transient climate simulations depend on the method employed in the analysis (Forster et al., 2016), meaning that further work is needed to evaluate the robustness of this relationship.

The simulated proportion of heat in the ocean for some models shows markedly different results depending on the used metric for total heat content in the climate system (Figure 4a). The different heat partition is caused by the discrepancies between estimates of N and EHC within each GCM simulation (Figure 2a), which are probably related to non-conservation terms in the simulated energy budget by each GCM as discussed in Hobbs et al. (2016). That is, small numerical inconsistencies, insufficient spin up time, or the amount of water leaving the LSM component at the bottom of the soil column, among others, may prevent the conservation of energy in GCM simulations (Sen Gupta et al., 2013; Hobbs et al., 2016; Séférian et al., 2016; Trenberth et al., 2016). We applied a simple drift-removal technique to each variable considered in this study in order to minimize the effect of possible non-conservation terms in our results (see Section 2). This method has shown good results in previous analyses including several CMIP5 experiments, although no perfect solution is available yet (Hobbs et al., 2016).

The low ground heat content achieved by the shallow LSM components (Figure 2b) alters the distribution of heat within models, mainly causing a higher proportion of heat stored in the ocean if considering EHC as metric for total heat content. This can be seen in a covariance analysis between OHC/EHC estimates and the depth of the LSM components in the CMIP5 ensemble (Figure S3). The shallow depth of the LSM components included in the CMIP5 GCMs limits the represented amount of continental heat storage within each simulation (Stevens et al., 2007; MacDougall et al., 2008; Cuesta-Valero et al., 2016; Hermoso de Mendoza et al., 2020), altering the GHC estimates and the obtained GHC/EHC and GHC/N ratios from the thirty CMIP5 GCMs analyzed here (Figures 2b, 4b and S3). Simulated OHC/N values, nevertheless, do not present such covariance

with the depth of the LSM component, nor the simulated proportion of heat in the atmosphere and the cryosphere (Figure S3 and S4). Surprisingly, the simulated CHC indicates significant covariance with the depth of the employed LSM component (red diamond in Figure 6), although this should be the result of the different subsurface volume within CMIP5 models. That is, deeper models tend to simulate more subsurface ice and GHC than shallower models, and therefore more heat can be used to thaw the larger mass of subsurface ice. This result suggests another limit to the representation of the EHI within GCM simulations, as the lack of a sufficient continental subsurface volume alters the simulated heat uptake by the subsurface ice masses. Nevertheless, further work is required to clarify this point.

The simulated cryosphere heat content and heat proportion are in better agreement with observations when ignoring the heat absorbed by terrestrial ice masses from the assessment, that is, considering only sea ice as cryosphere component (see results labelled as "only sea ice" in Table 2). The same can be said about the simulated proportion of heat in the ocean, which shows a reduction of 2% in the difference with observations if considering only sea ice as cryosphere (Table 2). This is caused by the lack of a representation of land ice in CMIP5 simulations, as only the simulated heat uptake by sea ice can be directly compared with observations, and the method used here to approximate the melting of land ice in the models is not accurate enough. Our approach considers snow changes in grid cells indicated as land ice by the models, but results show that this method markedly underestimates heat uptake in comparison with observations (Figure 3). Furthermore, the observed proportion of heat in the ocean yields different results if considering the whole cryosphere for estimating EHC or if considering only the change in sea ice volume (Table 2). Therefore, heat uptake by terrestrial ice sheets and glaciers is important to improve the simulated EHI and the partition of heat within the four climate subsystems. CMIP5 GCMs currently include modules representing ice sheets, but such model components were not activated for generating the CMIP5 simulations analyzed here, probably due to issues with computational resources and technical challenges of coupling the ice sheet grids with the rest of subsystems (Flato et al., 2013). New experiments are planned to assess the ability of the latest generation of GCMs to reproduce the ice sheets of Greenland and Antarctica within the sixth phase of the Coupled Model Intercomparison Project (CMIP6), including coupled atmosphere-ocean-ice-sheet simulations (Nowicki et al., 2016). Although these experiments are focused on understanding the contribution of ice sheets to sea-level rise, these simulations could be also useful to test if including land ice masses enhances the representation of the Earth heat inventory within GCMs, particularly the coupled experiments.

## 5 Conclusions

The ensemble of CMIP5 GCMs analyzed here overestimates the amount of heat stored in the ocean and underestimates the heat uptake by the cryosphere and the continental subsurface, while representing changes in atmosphere heat storage similar to observations. Models present a large inter-model spread of ocean heat content and total heat content in the system, probably related to the wide range of simulated responses to external forcing in these GCMs. The lack of an adequate representation of terrestrial ice masses and continental subsurface volume within CMIP5 models limits the amount of heat allocated within the cryosphere and the continental subsurface. The issue of heat conservation within complex numerical simulations also affects

the represented Earth heat inventory in the CMIP5 ensemble. Nevertheless, there is good agreement between simulated and observed atmosphere heat storage and heat uptake by changes in sea ice volume.

There are two main issues hindering the assessment of the EHI in CMIP5 models in comparison with observations, the non-conservation of energy in models and the markedly different amounts of simulated total heat content in the Earth system. Ocean heat storage is markedly high within the CMIP5 ensemble, presenting high inter-model variability. This causes a much higher Earth heat content in the models in comparison with observations. The different response of each model to the external forcing may be the cause for this large variability and high values of OHC and EHC, suggesting that simulations from the CMIP6 models may present an even larger spread in results, since Meehl et al. (2020) found a larger inter-model variability for estimates of Equilibrium Climate Sensitivity (ECS) in this new generation of models. Otherwise, the spread in Effective Radiative Forcing (ERF) is similar in CMIP5 and CMIP6 (Smith et al., 2020), and the simulated radiative imbalance seems to present smaller inter-model variability in the CMIP6 ensemble than in the CMIP5 ensemble (Wild, 2020). Therefore, a future assessment of the simulated EHI within the CMIP6 ensemble is required to determine the performance of the new generation of models. Regarding the non-conservation of energy within the models, Irving et al. (2020) have found that drifts in N and OHC are still markedly large in CMIP6 models, although the energy leakage within these models have improved in comparison with CMIP5 simulations. Nevertheless, such result indicates that an assessment of the represented EHI by CMIP6 models will encounter similar burdens and limitations as those in our analysis, including the need of applying a drift correction technique before evaluating the simulations.

The assessment of transient climate simulations in comparison with observations presented here indicates that deeper continental subsurfaces and some representation of terrestrial ice masses within GCMs are required to improve the simulated Earth heat inventory, as well as the associated phenomena relevant to society such as sea level rise or permafrost evolution. These issues will probably be present within the CMIP6 simulations, together with non-conservation of energy and drifts, but the comparison with observational references may help to mitigate this limitations in future generations of GCMs. For example, an extended sampling of the deepest part of the ocean will improve the observational estimate of OHC, and will provide a reference to evaluate deep heat uptake in GCMs, probably reducing the drift in these models (Irving et al., 2020; von Schuckmann et al., 2020). Local and regional measurements of the state of glaciers and ice sheets may help to parameterize the evolution of ice masses in individual grid cells. That is, a simplified parameterization of land ice masses based on tiling (Essery et al., 2003; Best et al., 2004) could be implemented. This strategy has been successful in representing vegetation functional types at sub-grid scales (Melton and Arora, 2014), and it has been proposed to improve the representation of permafrost in land surface model components (Beer, 2016). Additionally, expanding the global network of subsurface temperature profiles will improve the estimates of continental heat storage, mitigating the scarcity of measurements after 2000 CE and in the Southern Hemisphere (Cuesta-Valero et al., 2021).

Furthermore, the collaboration between the observational and modelling communities should be maintained and expanded to further advance our knowledge of key climate processes. Assessments of transient climate simulations based on observational estimates of important climate variables, like the analysis performed here, have been showing paths for improvement in climate modelling for a long time now. The analysis of CMIP6 simulations will allow for testing of the improvement of

advanced climate models in reproducing the evolution of climate change, but in order to maintain the progress in modelling and to enhance the understanding of the processes conforming the Grand Challenges of the World Climate Research Program (WCRP), the global network of observations must be maintained and expanded.

*Data availability.* CMIP5 simulations can be accessed at the dedicated website of the Earth System Grid Federation (https://esgf-node.llnl.gov/projects/cmip5/). Data from von Schuckmann et al. (2020) is available with DOI:https://doi.org/10.26050/WDCC/GCOS_EHI_EXP_v2, and data from Church et al. (2011) can be retrieved from the publication itself.

*Author contributions.* FJCV designed the study, analyzed the CMIP5 simulations, and produced all results and figures. All authors contributed to the interpretation and discussion of results. FJCV wrote the manuscript with continuous feedback from all authors.

*Competing interests.* The authors declare that they have no conflict of interest.

*Acknowledgements.* We acknowledge the World Climate Research Programme's Working Group on Coupled Modelling, which is responsible for CMIP, and we thank the climate modeling groups responsible for the model simulations used herein (listed in Table 1 of this paper) for producing and making available their model output. For CMIP the U.S. Department of Energy's Program for Climate Model Diagnosis and Intercomparison provides coordinating support and led development of software infrastructure in partnership with the Global Organiza-
405 tion for Earth System Science Portals. Part of the presented analysis was performed in the computational facilities provided by the Atlantic Computational Excellence Network (ACENET-Compute Canada). We thank two anonymous reviewers for their constructive feedback. We are grateful to Fiammetta Straneo and Susheel Adusumilli for their help to obtain and interpret the sea ice volume data. This work was supported by grants from the Natural Sciences and Engineering Research Council of Canada Discovery Grant (NSERC DG 140576948), the Canada Research Chairs Program (CRC 230687), and the Canada Foundation for Innovation (CFI) to H. Beltrami. H. Beltrami holds
Canada Research Chair in Climate Dynamics. A.G.G. and F.J.C.V. are funded by H. Beltrami's Canada Research Chair program, the School of Graduate Students at Memorial University of Newfoundland and the Research Office at St. Francis Xavier University.

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

**Table 1.** Variables from the CMIP5 archive employed to estimate the heat content within each climate subsystem by each GCM (Section 2). References for each GCM Historical experiment are also provided. All variables correspond with the r1i1p1 realization of the Historical experiment. A description of all listed variables can be found at the dedicated webpage of the Lawrence Livermore National Laboratory (LLNL, 2010).

| Model | Ocean | Land | Atmosphere | Cryosphere | TOA Imbalance | References |
|---|---|---|---|---|---|---|
| CCSM4 | so, thetao | mrlsl, tsl | hus, ps, ta, ua, va | mrfso, sic, sit, snw | rlut, rsdt, rsut | Gent et al. (2011) |
| CESM1-BGC | so, thetao | mrlsl, tsl | hus, ps, ta, ua, va | mrfso, sic, sit, snw | rlut, rsdt, rsut | Long et al. (2013) |
| CESM1-CAM5 | so, thetao | mrlsl, tsl | hus, ps, ta, ua, va | mrfso, sic, sit, snw | rlut, rsdt, rsut | Meehl et al. (2013) |
| CESM1-FASTCHEM | so, thetao | mrlsl, tsl | hus, ps, ta, ua, va | mrfso, sic, sit, snw | rlut, rsdt, rsut | Hurrell et al. (2013) |
| CESM1-WACCM | so, thetao | mrlsl, tsl | hus, ps, ta, ua, va | mrfso, sic, sit, snw | rlut, rsdt, rsut | Marsh et al. (2013) |
| NOR-ESM1-M | so, thetao | mrlsl, tsl | hus, ps, ta, ua, va | mrfso, sic, sit, snw | rlut, rsdt, rsut | Iversen et al. (2013) |
| NOR-ESM1-ME | so, thetao | mrlsl, tsl | hus, ps, ta, ua, va | mrfso, sic, sit, snw | rlut, rsdt, rsut | Tjiputra et al. (2013) |
| INM-CM4 | so, thetao | mrlsl, tsl | hus, ps, ta, ua, va | sic, sit, snw | rlut, rsdt, rsut | Volodin et al. (2010) |
| MIROC-ESM | so, thetao | mrlsl, tsl | hus, ps, ta, ua, va | mrfso, sic, sit, snw | rlut, rsdt, rsut | Watanabe et al. (2011) |
| MIROC-ESM-CHEM | so, thetao | mrlsl, tsl | hus, ps, ta, ua, va | mrfso, sic, sit, snw | rlut, rsdt, rsut | Watanabe et al. (2011) |
| MIROC5 | so, thetao | mrlsl, tsl | hus, ps, ta, ua, va | mrfso, sic, sit, snw | rlut, rsdt, rsut | Watanabe et al. (2010) |
| GFDL-CM3 | so, thetao | mrlsl, tsl | hus, ps, ta, ua, va | mrfso, sic, sit, snw | rlut, rsdt, rsut | Donner et al. (2011) |
| GFDL-ESM2G | so, thetao | mrlsl, tsl | hus, ps, ta, ua, va | mrfso, sic, sit, snw | rlut, rsdt, rsut | Dunne et al. (2012) |
| GFDL-ESM2M | so, thetao | mrlsl, tsl | hus, ps, ta, ua, va | mrfso, sic, sit, snw | rlut, rsdt, rsut | Dunne et al. (2012) |
| MRI-CGCM3 | so, thetao | mrso, tsl | hus, ps, ta, ua, va | mrfso, sic, sit, snw | rlut, rsdt, rsut | Yukimoto et al. (2012) |
| MRI-ESM1 | so, thetao | mrso, tsl | hus, ps, ta, ua, va | mrfso, sic, sit, snw | rlut, rsdt, rsut | Adachi et al. (2013) |
| MPI-ESM-LR | so, thetao | mrso, tsl | hus, ps, ta, ua, va | sic, sit, snw | rlut, rsdt, rsut | Giorgetta et al. (2013) |
| MPI-ESM-MR | so, thetao | mrso, tsl | hus, ps, ta, ua, va | sic, sit, snw | rlut, rsdt, rsut | Giorgetta et al. (2013) |
| MPI-ESM-P | so, thetao | mrso, tsl | hus, ps, ta, ua, va | sic, sit, snw | rlut, rsdt, rsut | Jungclaus et al. (2014) |
| CMCC-CM | so, thetao | mrso, tsl | hus, ps, ta, ua, va | sic, sit, snw | rlut, rsdt, rsut | Scoccimarro et al. (2011) |
| CMCC-CMS | so, thetao | mrso, tsl | hus, ps, ta, ua, va | sic, sit, snw | rlut, rsdt, rsut | Scoccimarro et al. (2011) |
| CANESM2 | so, thetao | mrlsl, tsl | hus, ps, ta, ua, va | mrfso, sic, sit, snw | rlut, rsdt, rsut | Arora et al. (2011) |
| IPSL-CM5A-LR | so, thetao | mrso, tsl | hus, ps, ta, ua, va | sic, sit | rlut, rsdt, rsut | Dufresne et al. (2013) |
| IPSL-CM5A-MR | so, thetao | mrso, tsl | hus, ps, ta, ua, va | sic, sit | rlut, rsdt, rsut | Dufresne et al. (2013) |
| IPSL-CM5B-LR | so, thetao | mrso, tsl | hus, ps, ta, ua, va | sic, sit | rlut, rsdt, rsut | Dufresne et al. (2013) |
| GISS-E2-H | so, thetao | mrlsl, tsl | hus, ps, ta, ua, va | mrfso, sic, sit, snw | rlut, rsdt, rsut | Miller et al. (2014) |
| GISS-E2-R | so, thetao | mrlsl, tsl | hus, ps, ta, ua, va | mrfso, sic, sit, snw | rlut, rsdt, rsut | Miller et al. (2014) |
| BCC-CSM1.1 | so, thetao | mrlsl, tsl | hus, ps, ta, ua, va | mrfso, sic, sit, snw | rlut, rsdt, rsut | Wu et al. (2014) |
| BCC-CSM1.1-M | so, thetao | mrlsl,tsl | hus, ps, ta, ua, va | mrfso, sic, sit, snw | rlut, rsdt, rsut | Wu et al. (2014) |
| HADGEM2-CC | so, thetao | mrlsl, tsl | hus, ps, ta, ua, va | mrfso, sic, sit, snw | rlut, rsdt, rsut | Collins et al. (2011) |

**Table 2.** Earth heat inventory and proportion of heat allocated in each climate subsystem from the thirty CMIP5 GCMs analyzed here (MMM), and observations from Church et al. (2011) (Ch11) and von Schuckmann et al. (2020) (vS20). Heat storage in ZJ, heat proportion in %.

| Magnitude | MMM | Ch11 | vS20 |
|---|---|---|---|
| N | $264 \pm 171$ | – | – |
| EHC | $256 \pm 177$ | 212 | $188 \pm 17$ |
| OHC | $247 \pm 172$ | 199 | $164 \pm 17$ |
| GHC | $5 \pm 9$ | 4 | $14 \pm 3$ |
| AHC | $2 \pm 2$ | 2 | $2.2 \pm 0.3$ |
| CHC | $2 \pm 3$ | 7 | $7 \pm 1$ |
| CHC (only sea ice) | $2 \pm 2$ | 2 | $2.5 \pm 0.2$ |
| OHC/N | $93 \pm 24$ | – | – |
| OHC/EHC | $96 \pm 4$ | 94 | $88 \pm 12$ |
| OHC/EHC (only sea ice) | $96 \pm 4$ | 96 | $90 \pm 12$ |
| GHC/N | $2 \pm 3$ | – | – |
| GHC/EHC | $2 \pm 3$ | 2 | $7 \pm 2$ |
| GHC/EHC (only sea ice) | $2 \pm 3$ | 2 | $7 \pm 2$ |
| AHC/N | $1.0 \pm 0.9$ | – | – |
| AHC/EHC | $1 \pm 1$ | 0.9 | $1.1 \pm 0.2$ |
| AHC/EHC (only sea ice) | $1 \pm 1$ | 0.9 | $1.1 \pm 0.2$ |
| CHC/N | $1 \pm 1$ | – | – |
| CHC/EHC | $1 \pm 1$ | 3 | $3.6 \pm 0.7$ |
| CHC/EHC (only sea ice) | $1 \pm 1$ | 1 | $1.2 \pm 0.2$ |

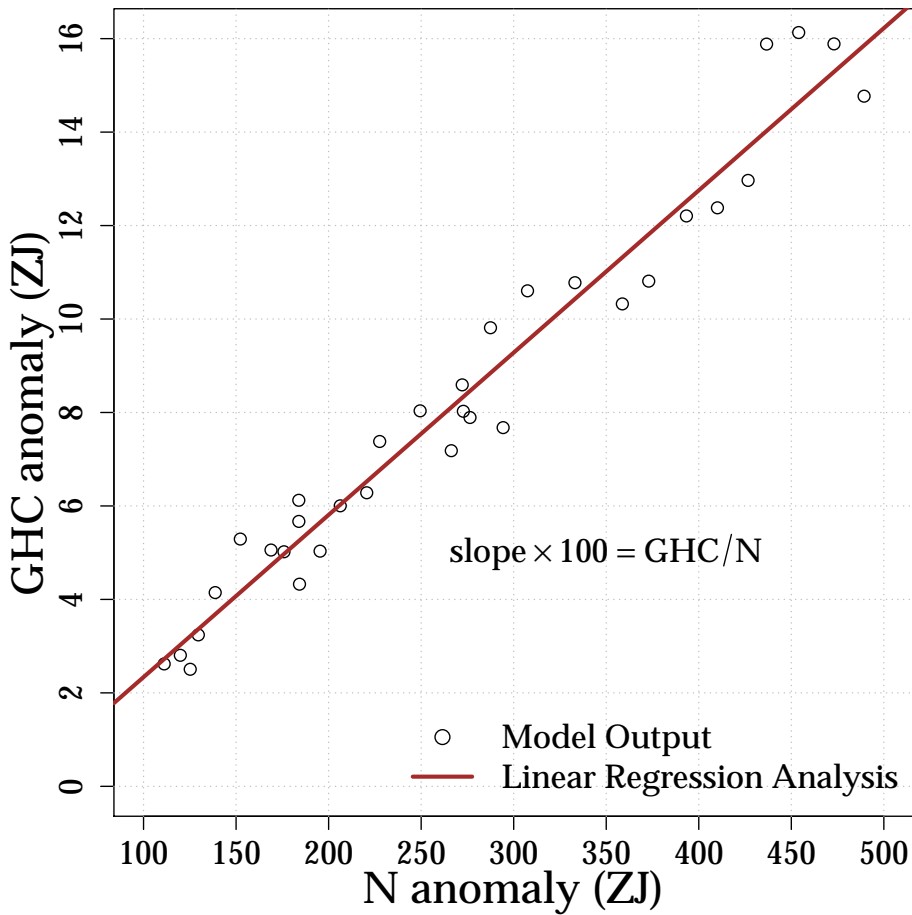

**Figure 1.** Example to illustrate the process to estimate heat proportions using data from the CCSM4 Historical simulation. In this case, the proportion of heat within the continental subsurface (GHC/N) is estimated as the slope from the linear regression analysis (solid line) between the simulated GHC and N anomalies (dots) for the period 1972-2005 CE multiplied by 100. The proportion of heat in the rest of climate subsystems is estimated replacing the GHC anomaly with the corresponding heat content anomaly. The EHC anomaly is also used as metric for the total heat content in the system by replacing the N anomaly in the regression analysis.

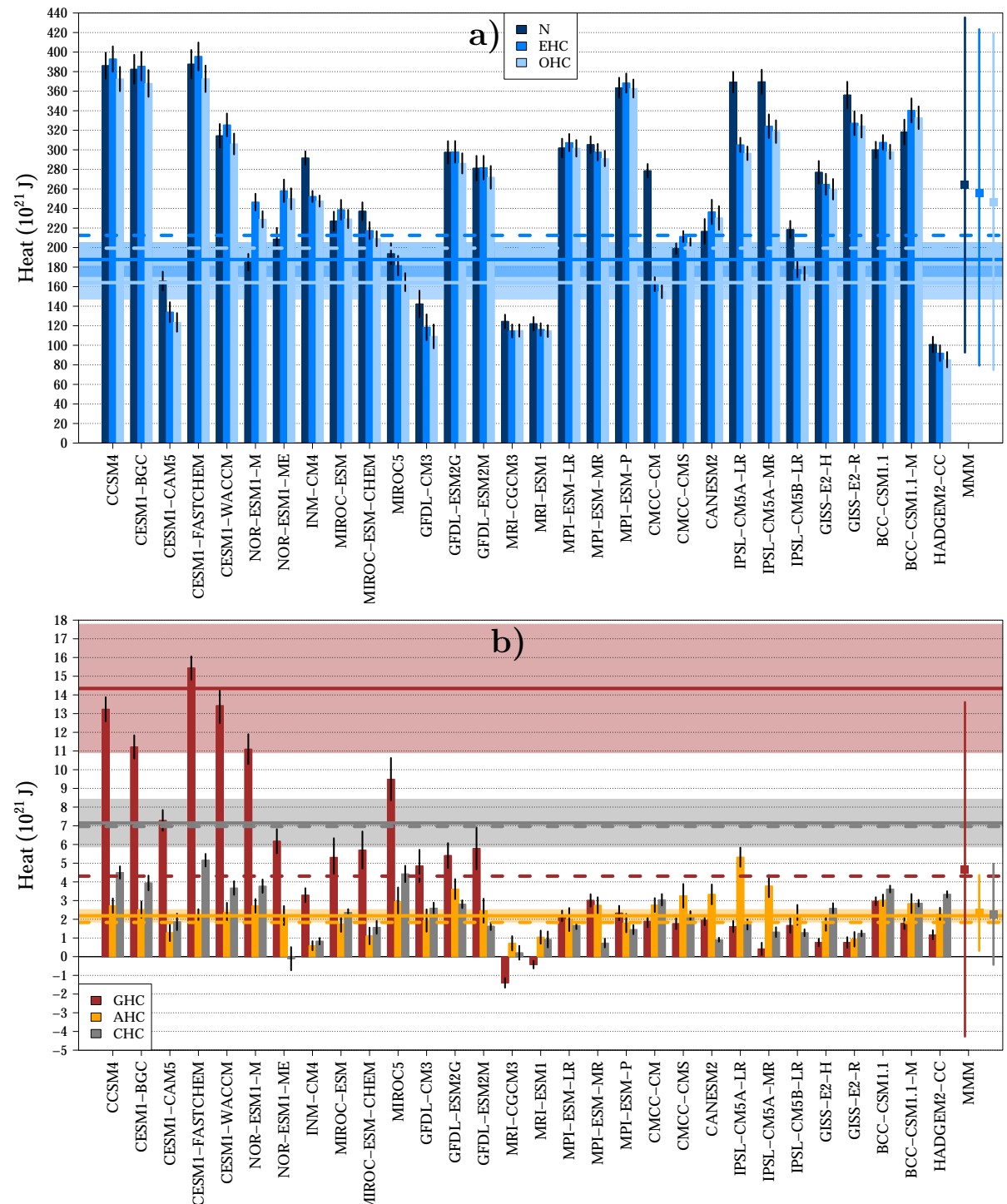

**Figure 2.** Simulated heat storage for 1972-2005 CE from thirty CMIP5 GCM Historical simulations. (upper panel) Results for N (dark blue bars), EHC (blue bars) and OHC (light blue bars). (bottom panel) Results for GHC (brown bars), AHC, (orange bars) and CHC (gray bars). Vertical black lines at the top of the bars indicate the 95% confidence interval for each model. Observations from von Schuckmann et al. (2020) are shown as solid horizontal lines and shadows (mean and 95% confidence intervals), and observations from Church et al. (2011) are displayed as dashed horizontal lines. Multimodel means and 95% confidence intervals are indicated in the right side of the panel (MMM).

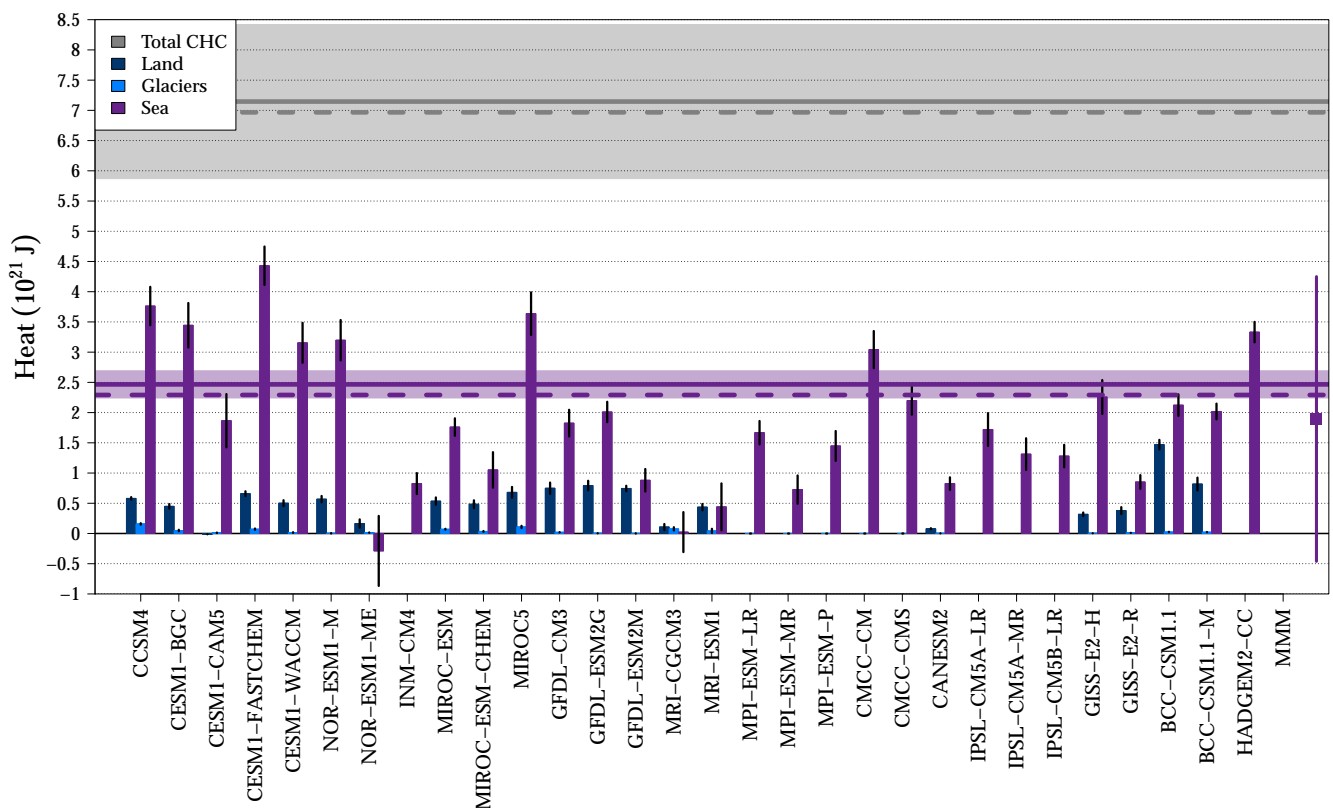

**Figure 3.** Simulated CHC for 1972-2005 CE. Dark blue bars indicate the heat uptake due to changes in subsurface ice mass, blue bars indicate the heat uptake due to changes in glacier mass, and purple bars indicate the heat uptake due to changes in sea ice volume (see Section 2 for details). Vertical black lines at the top of the bars indicate the 95% confidence interval for each model. The multimodel mean and 95% confidence interval for the heat uptake due to changes in sea ice volume are indicated in the right side of the panel (MMM). Observations from von Schuckmann et al. (2020) are shown as solid horizontal lines and shadows (means and 95% confidence intervals), and observations from Church et al. (2011) are displayed as dashed horizontal lines.

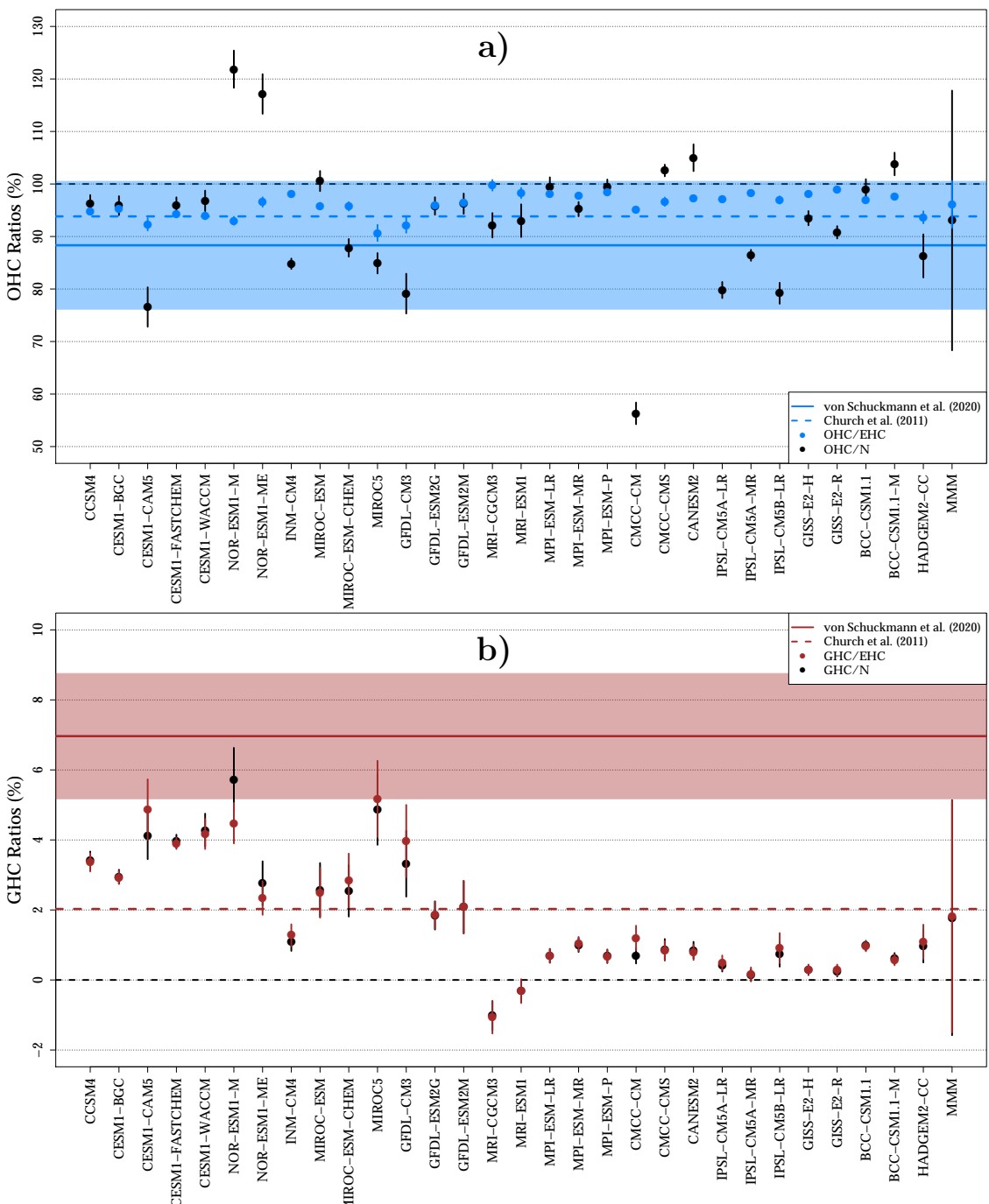

**Figure 4.** (a) Simulated proportion of heat within the ocean for the period 1972-2005 CE using EHC (blue dots) and N (black dots) as estimates of total heat content in the climate system. (b) Simulated proportion of heat within the continental subsurface for the period 1972-2005 CE using EHC (red dots) and N (black dots) as estimates of total heat content in the climate system. Observations from von Schuckmann et al. (2020) are shown as solid horizontal lines and shadows (means and 95% confidence intervals), and observations from Church et al. (2011) are displayed as dashed horizontal lines. Multimodel means and 95% confidence intervals are indicated in the right side of the panels (MMM). Black dashed lines indicate the 0% and 100% values.

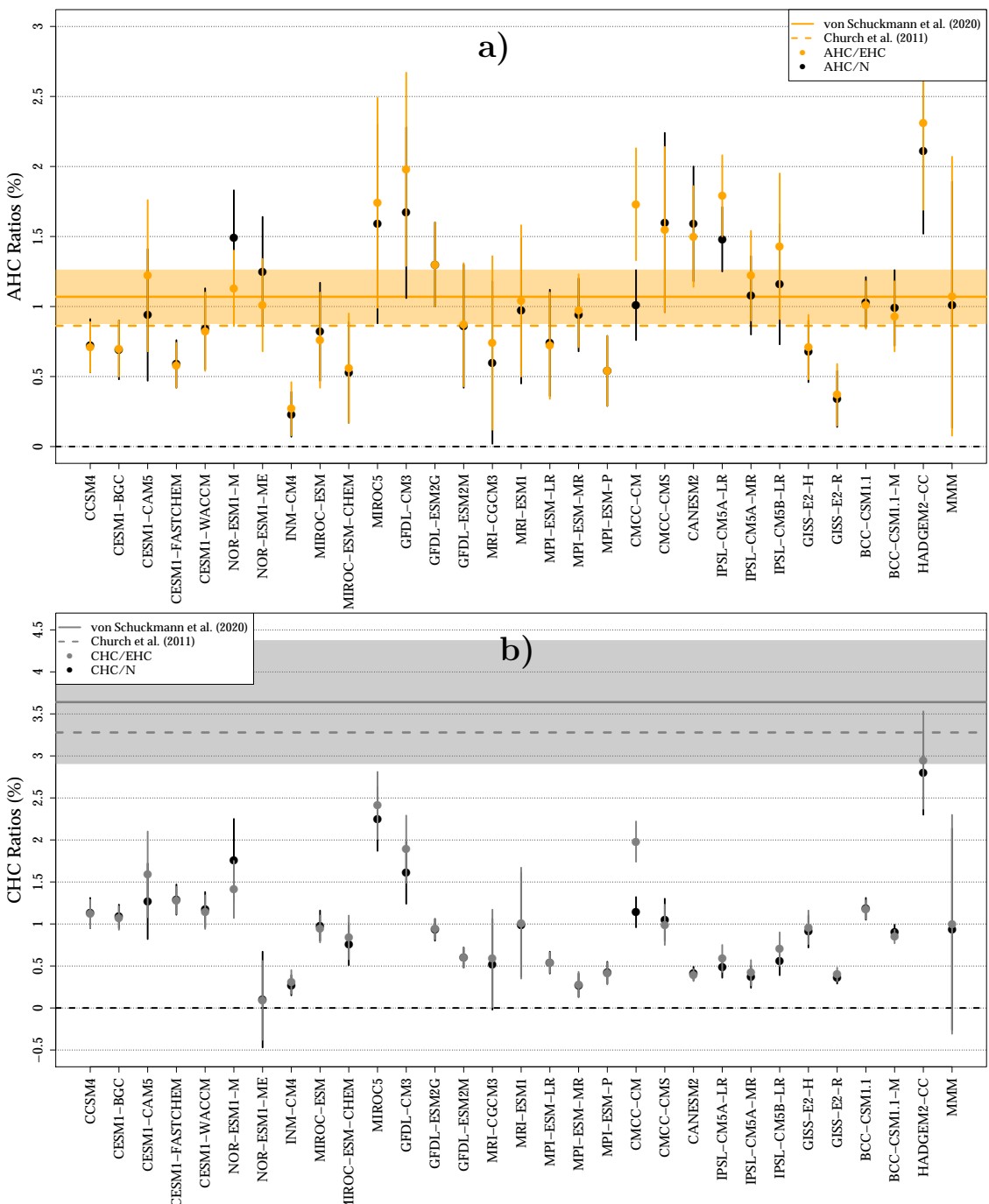

**Figure 5.** (a) Simulated proportion of heat within the atmosphere for the period 1972-2005 CE using EHC (orange dots) and N (black dots) as estimates of total heat content in the climate system. (b) Simulated proportion of heat within the continental subsurface for the period 1972-2005 CE using EHC (light blue dots) and N (black dots) as estimates of total heat content within the climate system. Observations from von Schuckmann et al. (2020) are shown as solid horizontal lines and shadows (means and 95% confidence intervals), and observations from Church et al. (2011) are displayed as dashed horizontal lines. Multimodel means and 95% confidence intervals are indicated in the right side of the panels (MMM). The black dashed line indicate the 0% value.

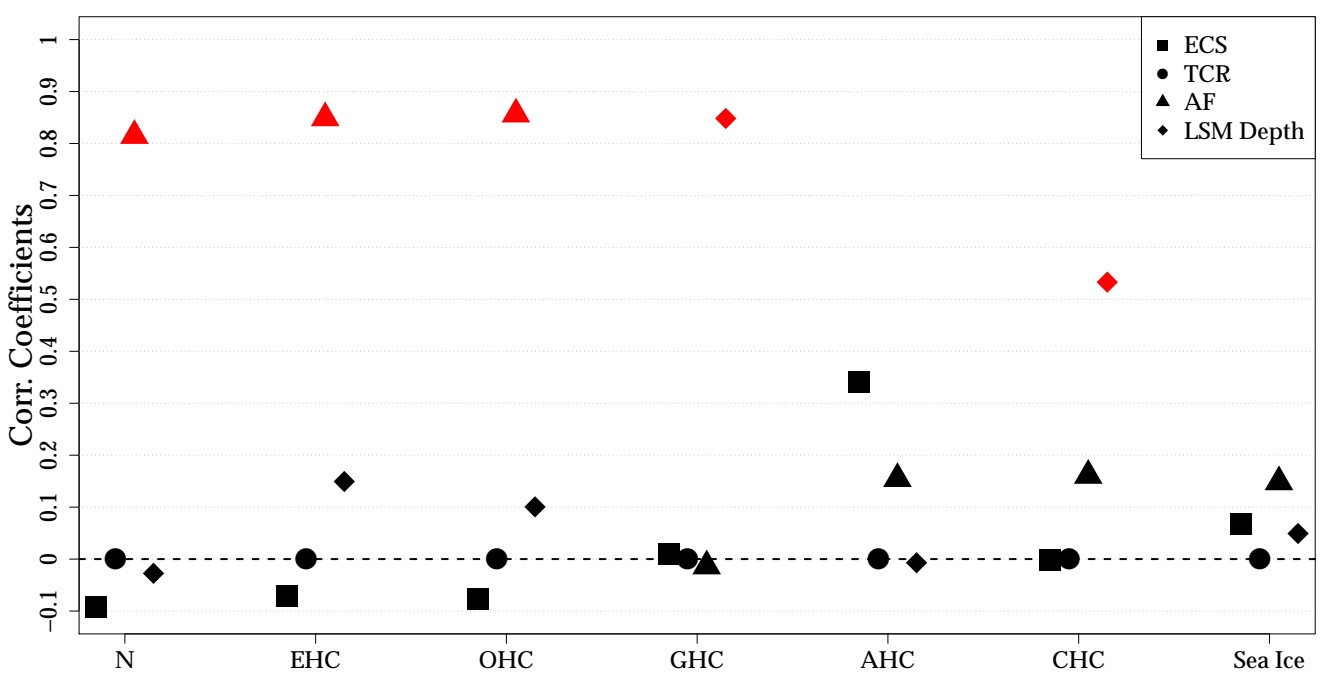

**Figure 6.** Correlation coefficients between the simulated Equilibrium Climate Sensitivity (ECS, squares), Transient Climate Response (TCR, circles), Adjusted Forcing (AF, triangles), depth of the LSM component (LSM Depth, diamonds) and the components of the Earth heat inventory. Results obtained by analyzing the eighteen models presenting estimates of ECS, TCR and AF in Forster et al. (2013). Red symbols indicate statistically significant results at the 95% confidence level using a Student's t-test. The dashed black horizontal line indicates zero values.