# Peer review of "First Assessment of the Earth Heat Inventory Within CMIP5 Historical Simulations"

_Earth System Dynamics, 2020_

## Referee Comment (RC1) · Anonymous Referee #1 · 3 Feb 2021

The article 'First Assessment of the Earth Heat Inventory Within CMIP5 Historical Simulations' provides an evaluation of the Earth heat inventory from climate model simulations, and assess the dissemination of heat storage distribution in the different Earth system components. The article is well written, timely and addresses a fundamental topic, and I recommend minor revision before publication following the different aspects provided below.

Comments :

L30-35 : The addition of more recent references would further support this part of the introduction, particularly while referring to outcomes of IPCC SROCC (and respective

chapters).

L84: This is not correct, as also observation-based products have been accounted for in their estimate: Wegener Center (WEGC) multisatellite RO data record, WEGC OPSv5.6 (Angerer et al., 2017), as well as its radiosonde (RS) data record derived from the highquality Vaisala sondes RS80/RS92/VS41, WEGC Vaisala (Ladstädter et al., 2015). Also, microwave sounding unit (MSU) data records (Mears andWentz, 2017) have been discussed, but have been finally excluded for the ensemble average used in the EHI. See Steiner et al. (2020) for references (https://doi.org/10.1175/JCLI-D-19-0998.1).

L97: This evaluation of ocean heat content is different from what is done by the observational community, where the integral of temperature anomalies is used instead of density integration. It would be interesting to understand why this approach is used here instead, and what the impact/difference between those different approaches are.

L.115-124: I recommend to consider the study of Steiner et al. (2020): https://doi.org/10.1175/JCLI-D-19-0998.1, 2020.

L.330-339: The conclusion could be extended a bit more, and draw a synthesis of all heat content components as discussed in the course of the article. More specific recommendations for future evolution, and knowledge gaps would further support the strength of the conclusion part. A specific element of discussion is also missing, i.e. on how the obtained results of this study further support the interpretation and future developments of climate models, and on how observation based and model based evaluations could seek strengthening of collaboration in the future to further advance on climate research topics, as well as on more robust and more robust potential for prediction validation – this is an essential element which should be addressed in this article. Finally, the consequences for climate models based on the outcomes, ie underestimates/overestimation of Earth system heat storage components should be commented as well (qualitatively in the conclusion, or as part of knowledge synthesis from

previous publications in the introduction part).

Minor:

Supplement Fig. S3: error in ref in figure caption (last sentence).

────────────────────────────

---

## Referee Comment (RC2) · Anonymous Referee #2 · 28 Feb 2021

Overall comments: The manuscript reports on an analysis of the ability of thirty CMIP5 models to simulate the distribution of heat within the Earth's energy reservoirs for the period 1972-2005. The manuscript calculates the total heat content as it is partitioned in the ocean, continental subsurface, cryosphere, and atmosphere and relates it to the TOA imbalance. The findings indicate that models overestimate the heat stored in the oceans and underestimate it in the continental and cryosphere. There are limitations in what the CMIP class models can represent – and that is reflected in the fact that the cryosphere component calculations do not account for the glaciers and ice sheet melting losses adequately and to some extent the continental component. So, the results have to be interpreted in that context. The manuscript is an important contribution towards understanding how well they represent the energy imbalance and the partitioning of heat energy within the earth system. Some comments listed below may be addressed to improve clarity.

Detailed comments: 1. Line 70: Does "consistently" here mean same period as observations? 2. Line 73: How about Von Schuckmann? What period are those results from? 3. Lines 73-74: How is this scaled? Linear or some other? If there is an increasing rate of storage, a linear may not be appropriate. This needs a little more elaboration. 4. Lines 127-130: This is not satisfactory and not really a good approximation. There really is no direct way to calculate the heat absorbed by melting glaciers / ice sheets in the CMIP5 class models - except using offline models. Perhaps this needs to be discussed in the paper a bit more. The way this is presented in subsequent sections, it appears to be estimated with the same level of confidence as the other components. 5. Line 151: What kind of trend from the piControl is removed? Linear or some other trend? 6. Lines 246-248: This is not really clear. There are no real outliers in figure 2. So if there is a point to be made about the discrepancy of specific models, they need to be identified (on fig 2) which they currently are not. 7. Lines 319-320: What does this mean? Leaving terrestrial cryosphere out of the ocean heat? Needs to be rewritten. 8. Lines 325-328: The ISMIP6 project (Nowicki et al., 2016) is a contribution to CMIP6 designed to quantify and understand the global sea level that arises due to past, present and future changes in the Greenland and Antarctic ice sheets, along with investigating ice sheet–climate feedbacks. It is not necessarily to reproduce ice Greenland and Antarctic sheets.

---

## Author Comment (AC1) · 22 Mar 2021

**Response to Reviewers Document for "First Assessment of the Earth Heat Inventory Within CMIP5 Historical Simulations" by Francisco José Cuesta-Valero, Almudena García-García, Hugo Beltrami, and Joel Finnis.**

**We thank the Reviewers for their thoughtful and constructive feedback.**

**This Response to Reviewers file provides a complete documentation of the changes made in response to each individual Reviewer's comment.**
**Reviewers' comments are shown in plain text. Author responses are shown in bold blue text.**

Reviewer 1

The article 'First Assessment of the Earth Heat Inventory Within CMIP5 Historical Simulations' provides an evaluation of the Earth heat inventory from climate model simulations, and assess the dissemination of heat storage distribution in the different Earth system components. The article is well written, timely and addresses a fundamental topic, and I recommend minor revision before publication following the different aspects provided below.

Comments:

L30-35: The addition of more recent references would further support this part of the introduction, particularly while referring to outcomes of IPCC SROCC (and respective chapters).

**More recent references have been added, including the IPCC SROCC.**

L84: This is not correct, as also observation-based products have been accounted for in their estimate: Wegener Center (WEGC) multisatellite RO data record, WEGC OPSv5.6 (Angerer et al., 2017), as well as its radiosonde (RS) data record derived from the highquality Vaisala sondes RS80/RS92/VS41, WEGC Vaisala (Ladstädter et al., 2015). Also, microwave sounding unit (MSU) data records (Mears and Wentz, 2017) have been discussed, but have been finally excluded for the ensemble average used in the EHI. See Steiner et al. (2020) for references (https://doi.org/10.1175/JCLI-D-19-0998.1).

**Thank you for noticing this. We have modified the text to reflect this comment.**

L97: This evaluation of ocean heat content is different from what is done by the observational community, where the integral of temperature anomalies is used instead of density integration. It would be interesting to understand why this approach is used

here instead, and what the impact/difference between those different approaches are.

**We use all available information from climate simulations to produce estimates of heat content as comprehensive as possible. These approaches do not always coincide with the methods employed to derive heat content estimates from observations, as the information available for each case is different. For example, estimates of ground heat content within CMIP5 simulations take into account the simulated water and ice content in the subsurface, which cannot be implemented in observation-based continental heat storage estimates. In the case of the ocean heat content, we concluded that using both temperature and salinity profiles constitutes a more comprehensive approach to estimate OHC changes, although such an approach is challenging to implement in global ocean observations due to the lack of salinity measurements. In any case, we performed additional OHC estimates integrating only simulated sea temperature profiles, reaching similar conclusions (Figure 1 in this document). We have included this figure as supplementary information, accompanied by a brief explanation on the text.**

L.115-124: I recommend to consider the study of Steiner et al. (2020): https://doi.org/10.1175/JCLI-D-19-0998.1, 2020.

**We appreciate the link to this recent research; however, we aren't sure there is a clear connection between Steiner et al. (2020; estimates temperature trends at different heights from a number of observational datasets), and the indicated lines of our text, which describe the method applied to estimate the atmospheric heat content from CMIP5 simulations. We have interpreted this as an indication that our wording was not clear, thus we have reworded the original text to improve its clarity.**

L.330-339: The conclusion could be extended a bit more, and draw a synthesis of all heat content components as discussed in the course of the article. More specific

recommendations for future evolution, and knowledge gaps would further support the strength of the conclusion part. A specific element of discussion is also missing, i.e. on how the obtained results of this study further support the interpretation and future developments of climate models, and on how observation based and model based evaluations could seek strengthening of collaboration in the future to further advance on climate research topics, as well as on more robust and more robust potential for prediction validation – this is an essential element which should be addressed in this article. Finally, the consequences for climate models based on the outcomes, i.e. underestimates/overestimation of Earth system heat storage components should be commented as well (qualitatively in the conclusion, or as part of knowledge synthesis from previous publications in the introduction part).

**We have expanded the Conclusions section in the new version of the manuscript addressing the points raised by the reviewer.**

Minor:

Supplement Fig. S3: error in ref in figure caption (last sentence).

**We have corrected this in the new version of the article.**
* * *
[Figure]

**Fig. 1.** Simulated change in OHC from integrating temperature and salinity profiles (blue) and from integrating temperature profiles alone (light blue). Observations indicated as horizontal lines and shadows.

---

## Author Comment (AC2) · 22 Mar 2021

**Response to Reviewers Document for "First Assessment of the Earth Heat Inventory Within CMIP5 Historical Simulations" by Francisco José Cuesta-Valero, Almudena García-García, Hugo Beltrami, and Joel Finnis.**

**We thank the Reviewers for their thoughtful and constructive feedback.**

**This Response to Reviewers file provides a complete documentation of the changes made in response to each individual Reviewer's comment.**

**Reviewers' comments are shown in plain text. Author responses are shown in bold blue text.**

Reviewer 2

Overall comments: The manuscript reports on an analysis of the ability of thirty CMIP5 models to simulate the distribution of heat within the Earth's energy reservoirs for the period 1972-2005. The manuscript calculates the total heat content as it is partitioned in the ocean, continental subsurface, cryosphere, and atmosphere and relates it to the TOA imbalance. The findings indicate that models overestimate the heat stored in the oceans and underestimate it in the continental and cryosphere. There are limitations in what the CMIP class models can represent – and that is reflected in the fact that the cryosphere component calculations do not account for the glaciers and ice sheet melting losses adequately and to some extent the continental component. So, the results have to be interpreted in that context. The manuscript is an important contribution towards understanding how well they represent the energy imbalance and the partitioning of heat energy within the earth system. Some comments listed below may be addressed to improve clarity.

Detailed comments:

1. Line 70: Does "consistently" here mean same period as observations?

**Exactly. Maybe that was not clear on the text, thus we have changed "consistently" by "in common" in the new version of the manuscript.**

2. Line 73: How about Von Schuckmann? What period are those results from?

**We have added details about the temporal period of results in von Schuckmann et al. (2020).**

3. Lines 73-74: How is this scaled? Linear or some other? If there is an increasing rate of storage, a linear may not be appropriate. This needs a little more elaboration.

**We have scaled the results linearly. Although the increasing in heat content with time in each climate subsystem is not typically linear, the period of interest is just three year shorter than the period examined in Church et al. (2011). This means that we reduce the original heat content in each subsystem by 8%, which is much smaller than the spread in both models and von Schuckmann et al. (2020). Therefore, we are confident that the possible error due to scaling would not change the conclusions of this manuscript, as an increase of 8% in the Ch11 column in Table 2 does not change the comparison with CMIP5 simulations and with von Schuckmann et al. (2020). In any case, we have explicitly stated that the scaling is linear in the new version of the manuscript to avoid confusion.**

4. Lines 127-130: This is not satisfactory and not really a good approximation. There really is no direct way to calculate the heat absorbed by melting glaciers / ice sheets in the CMIP5 class models - except using offline models. Perhaps this needs to be discussed in the paper a bit more. The way this is presented in subsequent sections, it appears to be estimated with the same level of confidence as the other components.

**Indeed, this is not an adequate method to estimate the contribution of ice sheets and glaciers to the cryosphere heat content. And we indicated that at several locations on the text. Nevertheless, this approach allows us to illustrate the need of including at least a first order representation of these ice masses in the simulations, since even considering our inadequate approach, CMIP5 simulations markedly underestimate the contribution of terrestrial ice masses. We have included a more clear statement on the new version of the text.**

5. Line 151: What kind of trend from the piControl is removed? Linear or some other trend?

**We use a linear trend for drift correction. The indicated line did not specifically say so, thus we have changed this line in the new version of the manuscript to clarify this point.**

6. Lines 246-248: This is not really clear. There are no real outliers in figure 2. So if there is a point to be made about the discrepancy of specific models, they need to be identified (on fig 2) which they currently are not.

**We refer here to the differences between N and EHC estimates shown in Figure 2a. Indeed, there are no outliers, but it can be seen that not all models present similar N and EHC values. Some models present large differences between both terms, which alters the estimated proportion of heat in the ocean if considering N or EHC as metric for the total heat content in the Earth system. As suggested, we have indicated the models obtaining excessively different N-EHC differences in Figure 2a, as well as models showing excessively high and low OHC/N in the new version of the text.**

7. Lines 319-320: What does this mean? Leaving terrestrial cryosphere out of the ocean heat? Needs to be rewritten.

**This means that observations and simulations of the proportion of heat in the ocean are in better agreement when the contribution from the cryosphere to EHC is considered as only sea ice melting. This is related to the lack of a representation of terrestrial ice masses in CMIP5 simulations, and how this biases the comparison with observations (see comment 4 above). We have clarified this point in the new version of the manuscript.**

8. Lines 325-328: The ISMIP6 project (Nowicki et al., 2016) is a contribution to CMIP6 designed to quantify and understand the global sea level that arises due to past, present and future changes in the Greenland and Antarctic ice sheets, along with investigating ice sheet–climate feedbacks. It is not necessarily to reproduce ice Greenland and Antarctic sheets.

**The reviewer is right, the main focus of the ISMIP6 is sea level rise. Nevertheless, one of the objectives of the protocols described in Nowicki et al. (2016) is to obtain an ensemble of simulations from fully coupled atmosphere-ocean-**

**ice-sheet frameworks (Section 3.2 in Nowicki et al., 2016). This approach would allow to estimate mass loses in both ice sheets, thus allowing to estimate the heat uptake by ice sheets with a coupled model configuration, typical in AOGCM experiments. We have clarified this on the text.**